# Changes in Perception of Illness during Cardiac Rehabilitation Programme among Patients with Acute Coronary Syndrome: A Longitudinal Study

**DOI:** 10.3390/healthcare11030311

**Published:** 2023-01-19

**Authors:** Sukhbeer Kaur Darsin Singh, Fatimah Binti Ahmedy, Abqariyah Binti Yahya Ahmad Noor, Khatijah Lim Abdullah, Imran Zainal Abidin, Anwar Bin Suhaimi

**Affiliations:** 1Department of Nursing, Faculty of Medicine and Health Sciences, Universiti Malaysia Sabah, Kota Kinabalu 88400, Sabah, Malaysia; 2Department of Nursing Science, Faculty of Medicine, Universiti of Malaya, Kuala Lumpur 50603, Wilayah Persekutuan, Malaysia; 3Department of Medical Education, Faculty of Medicine and Health Sciences, Universiti Malaysia Sabah, Kota Kinabalu 88400, Sabah, Malaysia; 4Department of Social and Preventive Medicine, Faculty of Medicine, Universiti of Malaya, Kuala Lumpur 50603, Wilayah Persekutuan, Malaysia; 5Department of Nursing, School of Medical and Life Sciences, Sunway University, Subang Jaya 47500, Selangor, Malaysia; 6Department of Nursing, Faculty of Medicine, Universiti Kebangsaan Malaysia, Bangi 43600, Selangor, Malaysia; 7Deparment of Medicine, Faculty of Medicine, Universiti of Malaya, Kuala Lumpur 50603, Wilayah Persekutuan, Malaysia; 8Department of Rehabilitation Medicine, Faculty of Medicine, Universiti of Malaya, Kuala Lumpur 50603, Wilayah Persekutuan, Malaysia

**Keywords:** acute coronary syndrome (ACS), cardiac rehabilitation programme, Brief Illness Perception Questionnaire, longitudinal study

## Abstract

Little is known about the changes in perception of illness among patients with the acute coronary syndrome (ACS) during cardiac rehabilitation programme (CRP). The purpose of this study is to determine changes in perception of illness with ACS patients during CRP to evaluate the association of patients’ characteristics with the perception of illness at the end of Phase II of CRP. A descriptive longitudinal study was conducted among 450 patients who attended 8-weeks of Phase II CRP at 2 public hospitals in Malaysia and perception of illness was assessed using Brief Illness Perception Questionnaire (BIPQ). The assessment was conducted before Phase II (T0), during the 4th session (T1), and at the end of right after the 8th session (T2). One-way repeated measures of ANOVA analysed the changes of perception at T1 and T2 while logistic regression analysis evaluated the association of patients’ characteristics with the perception of illness at T2. Perception of illness changed during and after CRP from T0 to T1, and T1 to T2 (*p* < 0.001). The patient viewed ACS as an illness that changed from being more acute to a chronic condition as the sessions progressed. Previous history of acute myocardial infarction (OR = 2.380, 95% CI 1.46, 5.49) and angioplasty intervention were both found to be associated with the perception of illness (OR = 3.857, 95% CI 1.55, 9.61). Perception of illness changed during CRP and these changes are associated with patients’ previous history of cardiac events. Phase II can be viewed as the second window of opportunity for healthcare professionals to intervene early in modifying the perception of illness.

## 1. Introduction

Perception of illness is defined as the personal hope for individuals in combating their illnesses and includes their beliefs like the illness’ complaints, the course of the disease, and the effects and the extent to which the illness is controllable either by self-care or medical care [1]. Illness beliefs are closely related to the Common Sense Model (CSM) of Self-Regulation by Leventhal, Philips, and Burns (2016), [1] which discussed on the processes by which patients become aware of a health threat, plan affective responses to the threat, formulate perceptions of the threat and potential treatment actions, create action plans for addressing the threat, and integrate continuous feedback on action plan efficacy and threat-progression.

Leventhal and colleagues (1997) described five components of these illness representations as identity; the label or name was given to the condition and the symptoms that ‘appear’ with it. The first component is ‘Identity’ the label the person uses to describe the illness and symptoms. The second component is ‘Consequences’ emphasizes on individual beliefs on the consequences of the condition and how this will impact them physically and socially. The third component is ‘Cause’ where individualistic ideas on the perceived cause of the condition, may not be completely biomedically accurate. The fourth component is ‘Timeline’, acting as the predictive belief about how long the condition might last. The fifth component, ‘Control’ is beliefs on whether the condition can be cured or kept under control, and the degree to which the individual plays a part in achieving this state. With the integration of CSM, individuals with ACS may obtain new information about their conditions and evaluate their attempts to moderate, cure or cope with the effects of these conditions. Subsequently, new representations of illness beliefs are formed and developed based on their experiences going through the course of the disease [2]. These representations are closely related to the coping behaviours during CRP.

Cardiovascular disease (CVDs) is observed to rank on top of non-communicable diseases (NCDs) worldwide [3]. In 2011, The United Nations recognized NCDs including CVDs as a major threat to global health and set out an ambitious plan to reduce the effect and impact of these diseases across the globe. In the recent Health and Morbidity Survey 2019), coronary heart disease (CHD) has appeared as one of the leading causes of death in Malaysia. Illness perception on how patients with acute coronary syndrome (ACS) perceived values towards their illness may enhance or demotivate them during CRP.

ACS is commonly used to explain a pattern of manifestations that leads to CHD. Guidelines by (American Heart Association (AHA) and American College of Cardiology (ACC) in 2014) [4] and the (European Society of Cardiology (ESC) in 2017) [5] have clearly stipulated that the diagnosis of ST-Segment Elevation Myocardial Infarction (STEMI) is made based on the clinical presentations of ACS signs and symptoms as well as findings from the ECG. The Clinical Practice Guidelines Management of acute STEMI published in (2019) by the Ministry of Health Malaysia has also defined diagnostic criteria of ACS.

Next to pharmacological therapies and interventional cardiology, CRP is widely recognized as an essential treatment to the care of patients with ACS. CRP is proven to be a valuable part of the holistic management approach for CAD and serves as a secondary prevention measure [6]. CRP aims to optimize the physical functionality and reduce the recurrence of major cardiac and cerebrovascular events [7]. CRP usually commences when patients with ACS are stable and discharged from specialised cardiac or coronary care unit prior to leaving the hospital. CRP is a rehabilitation-led and medically supervised program designed to improve the cardiovascular fitness of patients with ACS. The CRP is multilayered interventional programme that optimises these patients on physical, psychological and social functioning aspects and comprises multidisciplinary approaches with various processes emphasizing on physical exercise, changes in behaviors aimed at healthier lifestyles, control of risk factors, and psychological interventions, with the main reason for delaying the progression of the underlying cardiovascular disease [8].

It has been shown that the CRP is widely underutilized from the overall participation rates in recent decades of about 40% [9]. Commonly, after any illnesses, the anticipated cognitive responses such as illness perception influences individual’s responses to the illness, self-management, treatment adherence, and coping strategies [10]. The attendance rate for cardiac patients in the CRP of a tertiary center in at the local settings was only 12% from the total annual admission rate of 7000 patients (Hospital A, Annual Report, 2016). Till date, research studies on illness perception and its relation to CRP among ACS patients have not been conducted in any of the Asian countries.

In majority of tertiary hospital settings in this region, the CRP is conducted in three phases and the term is utilized to describe the changes in time frame following a coronary event with specific aims and interventions in each phase [11]. Phase I of CRP takes place in the hospital setting at 24–48 h after a non-complicated coronary event, or post-cardiac surgery and continues for approximately 6 weeks. Phase II is usually considered at 6 weeks post-discharge and conducted at hospital facilities that provide cardiac rehabilitation follow-up and CRP. The minimal duration of the Phase II is 8 weeks and may increase up to 12 weeks. The final Phase III is known as the long-term phase and emphasizes on self-care management with regular follow-up intervals conducted every 2–6 months until 18–24 months post-cardiac event. This article is presenting the findings of a one-year longitudinal survey on illness perception changes from the end of Phase I to the end of Phase II among patients with ACS in CRP, and association between illness perception and patients’ clinical characteristics.

## 2. Materials and Methods

### 2.1. Study Design

The study was conducted as a descriptive longitudinal study for 1 year period beginning the January 2019 until December 2019 at two tertiary hospitals in the central part of Peninsular Malaysia.

### 2.2. Study Setting

The first hospital, Hospital A is governed under the Ministry of Higher Education (MOHE) Malaysia and the second hospital, Hospital B is public hospital under the Ministry of Health (MOH) Malaysia. The former hospital is the center of providing training, education and research related to cardiac rehabilitation since 2007 and the latter is the main cardiac referral center under the MOH in Peninsular Malaysia with extensive CRP services. In total, both hospitals allocated 130 active beds patients with CHD scattered in coronary care units, cardiac rehabilitation wards, and cardiac interventional wards. Inclusion criteria were patients enrolled into CRP, 18 years of age, ability to provide consent and speak both English and Malay languages. Those with terminal illnesses other than cardiac-related and presence of neurological deficits were excluded from the survey.

The implementation and structure of the CRP in both hospitals are similar (Table 1)**.** The sessions are conducted by in-house rehabilitation specialists, designated physiotherapists and occupational therapists, nurses, pharmacists and dieticians at Department of Rehabilitation Medicine of respective hospitals. Eligible participants were counseled and consented for the study at the end of Phase I and followed up during and after Phase II ended.

### 2.3. Participants

The sample size was calculated with Open-Source Epidemiologic Statistics for Public Health Info version 3.01 (2013) by comparing two means based on the power of 80% and confidence interval of 95% with the ratio of sample size of 1:1. An additional 20% for non- response rate to account for the missing values were added as demonstrated in previous study by Shanmugasegaram, Gagliese [12]. The minimum required sample size for the survey was 450 patients.

### 2.4. Study Instrument

The data collection form consisted of demographic data, clinical information and Brief Illness Perception Questionnaires (BIPQ) [13]. BIPQ is a 9-item self-rated questionnaire using 11-point Likert Scale ranging from 0–10 where higher scores indicate stronger perceptions along the specified dimensions in the first 8 items. Five of the items assess cognitive illness representations: consequences (Item 1) on the expected outcome of the illness, timeline (Item 2) on the duration that the patient believes that the illness will last, personal control (Item 3) on the extent that the patients believe they can control from the illness, treatment control (Item 4) where patients perceive that the treatment shall ace them to recover from their illness, and identity (Item 5) whereby patients view symptoms as part of their disease manifestations.

Two items assess emotional representations which are concern (item 6) and emotions (Item 8) that explore how emotionally patients are affected by the disease. One item assesses illness comprehensibility (Item 7) on the extent of patients’ understanding on the illness. The final item 9 is an open-ended response that asks patients on the three most important factors in illness. Responses to the causal items can be grouped into categories such as stress, lifestyle, and hereditary. The questionnaires were prepared in dual language (English and Malay) to enable patients to respond based on their preferred language. The English version of the instrument was forward and backward translated by a language expert to ensure semantic equivalence of the questionnaire and the content and face validity were obtained. The validity and internal consistency of the translated questionnaires was conducted among 40 patients and the Cronbach Alpha of 0.802 was obtained.

### 2.5. Study Procedure

The data collection started when patients were referred to CRP at Phase I and gradually during the Phase II. The baseline data collection (T0) was conducted during Phase I after eligible patients during their admission for an ACS event in the cardiac ward have consented. Patients were given 15 to 20 min to complete the BIPQ. During the Phase II of CRP, repeated self-assessment using BIPQ were conducted at the 4th session (T1) and 8th session (T2) at outpatient setting.

At baseline (T0), a total of 798 patients were found to be admitted with cardiac event in these two hospitals. Out of these, 600 patients fulfilled the eligibility criteria (400 from Hospital A and 200 from Hospital B). Hence, they were counseled for CRP enrolment and recruited with consent for the survey. However, at the end of Phase I (T0), 300 and 150 patients from Hospital A and Hospital B have completed the BIPQ prior discharged, making it a total of 450 correspondence at baseline. There was a dropout of 100 and 31 patients from Hospital A and Hospital B correspondingly, mainly due to these patients being uncontactable and declined for study participations at the end of the study. Eventually, a total of 319 patients completed the 8th session of Phase II in CRP (T2) (Figure 1).

### 2.6. Statistical Analysis

Data was analysed using the Statistical Package for Social Science (SPSS) version 25. Descriptive statistics including mean and standard deviation (SD) for continuous variables and frequency for categorical variables were explored for baseline data. The repeated measures of ANOVA were used to compare the data at different point of times from T0, T1 and T2. Binary logistic regression was used to measure the patient’s illness perception at T2.

## 3. Results

### 3.1. Baseline Demographics and Clinical Characteristics

Descriptive analysis was conducted for eligible 450 patients evaluated at baseline (T0). The mean age of the patients was 54.88 (7.51) and there were more males (Table 2). Malays were predominant ethnic, and majority were married. Less than a quarter had tertiary education (22.9%) and less than half were employed (45.7%). The commonest ACS event occurred was angina (87.8%) and diabetes mellitus ranked the highest (56.7%) of all comorbidities (Table 3). Less than quarter of them attended CRP (27.8%) and actively smoke.

### 3.2. Changes in Illness Perception

The overall mean score for illness perception was 43.9 (9.9) at T0, 53.8 (5.6) at T1 and 45.2 (11.5) at T2 (Table 4). There was an increase in the overall illness perception mean score from baseline (T0) to 4th session (T1) but subsequently decreased at 8th session (T2). The changes for individual items (Item 1–8) of BIPQ and One-way repeated measures of ANOVA statistically displayed that these 8 items have changed significantly from T0, T1 and T2 overtime. Patients perceived that treatment could help in controlling their cardiac disease, perceived fewer consequences on how cardiac disease could affect their life, perceived longer duration on the progress of ACS, increased concern over ACS as well as emotional response. Patient perceived more symptoms but better control towards ACS and perceived better understanding on ACS. For open-ended question of Item 9, majority of the patients emphasised on stress, lack of exercise, feeling tired, and weakness as their main concerns related to ACS.

### 3.3. Association between Clinical Characteristics and Illness Perception

Patients with previous cardiac history of AMI were 3.0 times more likely to have higher illness perception (OR = 2.380, 95% CI = 1.46, 5.49) compared to those without previous AMI. On the other hand, patients that had angioplasty with stent insertion and open-heart surgery were 4.0 times (OR = 3.857, 95% CI = 1.55, 9.61) and 2.2 times (OR = 2.239, 95% CI = 1.27, 3.96) more likely to have higher illness perception respectively compared to those without these cardiac procedures. Meanwhile, the analysis has shown that those with hypercholesterolemia have a higher illness perception (OR = 0.417, 95% CI = 0.18, 0.94) compared to those without this risk factor (Table 5).

## 4. Discussion

The purpose of this study was to determine changes in the illness perception from the end of Phase I, midway and until the end of Phase II of participating in CRP. In addition, the study also evaluated presence of association between clinical characteristics and illness perception assessed at the end of Phase II. The results of our study showed that there were changes in the illness perception overtime covering the dimension of consequences, timeline, personal control, treatment control, identity, concern, understanding, and emotional response based on BIPQ. These findings were similar to a study by [14] that assessed on illness perception in 98 patients with epilepsy on the consequences and emotional representation (17.41 ± 5.22 and 21.73 ± 5.79 respectively) based on the Chinese Illness Perception Revised (CIPQ-R). Although the personal control and treatment control in our study was lower than the study by [15] because of their the utilization of CIPQ-R, but ours are relatively similar to the findings by [16].

There was an increase of mean score in individual 8 items of illness perception and based on the mean score from highest to the lowest attained, the sequence is as follows: consequences, timeline, personal control, treatment control, concern and understanding. This findings were similar to [16] among 158 participants with ischaemic heart disease to determine the changes of illness belief during participation in CRP. Our study findings were congruent to a study by [17,18] where there were significant differences in the mean scores of the consequences, timeline, and control/cure at the time of discharge indicating that patients had lower level of belief that their cardiac condition would cause serious consequences and may last a long time or indefinitely. Thus, patients in their study perceived that their illness as more chronic and less controllable over time, both through personal efforts and treatment.

Our study findings were incongruent with a cross-sectional study by [19] conducted among 93 individuals in the United Kingdom for identifying psychological barriers in attenders and poor/non-attenders during Phase II of CRP. Their findings showed that attenders had significantly higher scores on identity and consequences with no significant differences between attenders and poor/non-attenders on illness timeline and controllability. In their study, the Illness Perception Questionnaire (IPQ) tool was utilized, and this has more items compared to BIPQ, thus a measurement of tools may contribute to equivocal findings.

The findings in our study were comparable in the study by [20,21] where patients’ beliefs on suitability of CRP and medication adherence at the start of the program are especially important to medication adherence at 6 months. Medication adherence should be addressed early as part of the cardiac rehabilitation process as well as any mistaken beliefs about CRP. A systematic review and meta-analysis by [22] proved that the attendance at CRP was predicted by four dimensions of illness perceptions; patients with more positive identity, cure/control, consequences and coherence beliefs are more likely to attend CRP. Therefore, patients viewed their illness as more chronic and less controllable over a period of time from overtime.

CRP usually takes place from the acute phase of ACS in which illness perception are dynamically changing as a result in the changes in the treatment and disease status. We found that from baseline (T0) to the 4th session of Phase II (T1), patient perceived an increased sense of understanding of ACS, believed that it is going to take a longer time (timeline) as their illness become chronic, felt that their personal control increased with greater concern on their illness. They have also attributed more symptoms and concerns towards their illness and felt that being diagnosed with ACS have extremely affected them emotionally overtime. It can be concluded that patient view their illness from an acute to a more chronic phase and perceived their illness negatively with time. Patients felt that despite attending CRP, it might not improve their health status therefore the high drop-out rate among patients (Figure 1) from the initial of CRP to at the 8th session (T2).

Despite significant findings in all of the individual items of illness perception domains, patients perceived lessen sense in the treatment control, experienced decrease sense in the understanding, lessened emotional response, perceived shorter timeline, fewer symptoms and, consequences as well as less concern over their disease. Nonetheless, illness perception is still susceptible to change, thus providing a window of opportunity during which negative illness perceptions during midway of Phase II that are not in accordance with disease severity can be altered and positive outcomes at the end of Phase II can be facilitated. Currently, CRP is shown to be relatively cost-effectively and different healthcare practitioners can be trained adequately to deliver interventions aimed at changing illness perceptions of patients over a period of time [23].

The second objective of our study was to determine the association of illness perception with clinical characteristics at the end of Phase II of CRP. Our study has demonstrated previous cardiac history of AMI, underwent angioplasty with stent insertion, and open-heart surgery, as well as hypercholesterolemia were significantly associated with illness perception. We found no significant associations with sociodemographic factors. These results were incongruent with study findings by [24] where they found active smoking status, low socioeconomic status, younger age, and patients with conservative management (non-surgical cardiac history) completed fewer sessions of CRP. In our study, a smoking variable was also measured as part of the past clinical history characteristics, but we failed to show significance association.

The results of our study was also congruent with the findings from [25] in which the latter demonstrated that diagnosis following a coronary angiography had shown a prompt effect on patients’ illness perception and emotional response towards their diagnosed condition. A patient who received normal results had lower symptoms associated with their illness, thus it influenced their emotional response towards the illness. The association of illness perception with hypercholesterolemia in our study was comparable to the report by [26] whereby the non-attendees of CRP reported higher total illness perception scores and those who attributed their illness to non-modifiable factors were less likely to attend CRP. Our findings were also similar to the study by [19], indicating that individuals who perceived a greater number of symptoms manifestations after AMI are more likely to attend the CRP sessions. Post-cardiac surgery patients who have better understanding in their illness before the surgery engaged better and positively improved in their physical functions after cardiac surgery [27]. Our findings concurred with their results as we have shown that the history of open-heart surgery was significantly associated with illness perception at the end of Phase II of CRP.

### Study Limitations

The study has limitations despite been conducted at two hospital settings. There was some loss of data due to the high rate of drop out which may impact the statistical power and reduce the generalizability of the results. Further research is needed to conduct an interventional study that involve a larger random sample size, more hospitals participations and objective measurement in physical components such as the changes in their blood pressure and heart rate during their attendance in CRP.

## 5. Conclusions

The changes in the illness perception from Phase I to Phase II of CRP has changed the patients’ perception on how they view the cardiac event and gradually becoming more acceptable and able to cope with their illness. Perception of illness changed during CRP and these changes are associated with patients’ previous history of cardiac events. These perceptions are potentially modifiable and should be aimed as early as Phase of CRP. Phase II can be viewed as the second window of opportunity for healthcare professionals to intervene early in modifying the perception of illness arisen from maladaptive mechanism of coping. The evaluation described has the purpose of changing the perception and the management of the illness during cardiac rehabilitation process. Therefore, healthcare professionals involved in the CRP must actively collaborate to help patients perceived their illness towards positive manner for better outcomes.

## Figures and Tables

**Figure 1 healthcare-11-00311-f001:**
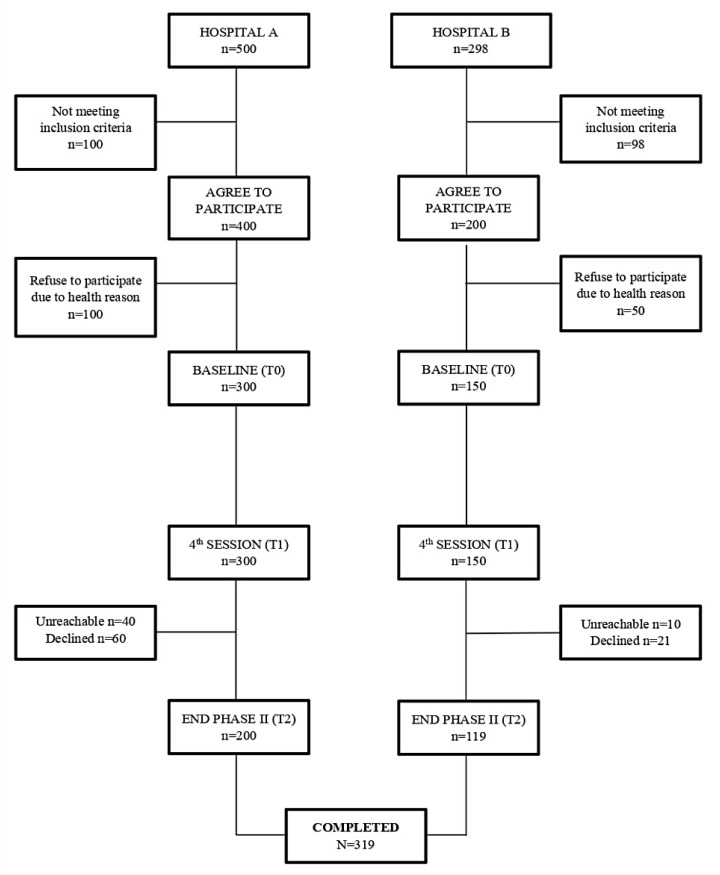
Study Flow Diagram.

**Table 1 healthcare-11-00311-t001:** Phase I and Phase II of Cardiac Rehabilitation Programs at selected study centers.

Phases of CRP	Activities
**Phase I**	**Education intervention on:** Introduction on CRP Anatomy and physiology of the heart Diet management Breathing exercises and isometric exercises
**Phase II**	**Education Intervention (as per Phase I) and including the following:** Structured and graded intensity physical exercises and activities for a duration of 60 min/session in 8 consecutive weeks: 10 min warm-up 10 min static bike with a gradual increase in intensity 20 min of walking 15–20 min treadmill Activities of daily living (ADL) Breathing exercise and stress relieve management

**Table 2 healthcare-11-00311-t002:** Demographics characteristics of the patients.

Characteristics		N = 450 (%)
**Age in years (mean, +/− SD)**		54.88, +/− 7.51
**Gender**	Male	337 (74.9%)
	Female	113 (25.1%)
**Ethnicity**	Malay	215 (47.8%)
	Chinese	155 (34.4%)
	Indian	57 (12.7%)
	Others	23 (5.1%)
**Educational level**	Primary	175 (38.9%)
	Secondary	172 (38.2%)
	Diploma	82 (18.2%)
	Degree	21 (4.7%)
**Marital status**	Married	356 (79.1%)
	Single	13 (2.9%)
	Divorced	58 (12.9%)
	Living with a partner	23 (5.1%)
**Employment status**	Government	110 (24.4%)
	Private	96 (21.3%)
	Retired	131 (29.1%)
	Not employed	113 (25.1%)

**Table 3 healthcare-11-00311-t003:** Clinical characteristics of the patients.

Characteristics		All N (%)	Male N (%)	Female N (%)	χ^2^ Value	*p* Value
*Previous cardiac history*						
Angina	Yes	395 (87.8)	294 (74.4)	101 (25.6)	0.361	0.548
	No	55 (12.2)	43 (78.2)	12 (21.8)		
AMI	Yes	341 (75.8)	253 (74.2)	88 (25.8)	0.362	0.547
	No	109 (24.2)	84 (77.1)	25 (22.9)		
Angioplasty with stent	Yes	393 (87.3)	299 (76.1)	94 (23.9)	2.346	0.141
	No	57 (12.7)	38 (66.7)	19 (33.3)		
Open-heart surgery	Yes	150 (33.3)	105 (70.0)	45 (30.0)	2.86	0.091
	No	300 (66.7)	232 (77.3)	68 (22.7)		
*Comorbidities*						
Diabetes Mellitus	Yes	255 (56.7)	196 (76.9)	59 (23.1)	1.219	0.27
	No	195 (43.3)	141 (72.3)	54 (27.7)		
Peripheral vascular disease	Yes	38 (8.4)	24 (63.2)	14 (36.8)	3.037	0.081
	No	412 (91.6)	313 (76.0)	99 (24.0)		
Stroke	Yes	26 (5.8)	22 (84.6)	4 (15.4)	1.388	0.239
	No	424 (94.2)	315 (74.3)	109 (25.7)		
*Cardiac risk factors*						
Current smoker	Yes	112 (24.9)	95 (84.8)	17 (15.2)	7.823	* 0.005
	No	338 (75.1)	242 (71.6)	96 (28.4)		
History of smoking	Yes	307 (68.2)	248 (80.8)	59 (19.2)	17.84	* 0.000
	No	143 (31.8)	89 (62.2)	54 (37.8)		
Hypercholesterolemia	Yes	289 (64.2)	214 (74.0)	75 (26.0)	0.303	0.582
	No	161 (35.8)	123 (76.4)	38 (23.6)		
Hypertension	Yes	325 (72.2)	243 (74.8)	82 (25.2)	0.009	0.925
	No	125 (27.8)	94 (93.6)	31 (24.8)		
Sedentary lifestyle	Yes	143 (31.8)	103 (72.0)	40 (35.9)	0.912	0.34
	No	307 (68.2)	234 (76.2)	73 (23.8)		
Involvement in cardiologist care	Yes	238 (52.9)	189 (79.4)	49 (20.6)	5.495	0.019
	No	212 (47.1)	148 (69.8)	64 (30.2)		
Attended any sort of CRP	Yes	125 (27.8)	95 (36.0)	30 (24.0)	0.114	0.809
	No	325 (72.2)	242 (74.5)	83 (25.5)		

* *p*-value significant at 0.005.

**Table 4 healthcare-11-00311-t004:** Changes of Illness Perception at T0, T1 and T2.

BIPQ Items	Score	Baseline (T0)	4th Session (T1)	8th Session (T2)	*df*	*F*	*p* Value *
	Range	Mean ± SD	Mean ± SD	Mean ± SD			
Consequences	0–10	5.9 ± 2.3	6.6 ± 1.6	5.2 ± 2.3	1.874	40.51	<0.001
Timeline	0–10	6.3 ± 2.0	7.5 ± 1.7	5.8 ± 2.3	1.85	61.86	<0.001
Personal Control	0–10	4.9 ± 2.0	7.0 ± 1.6	5.8 ± 2.3	1.828	86.55	<0.000
Treatment Control	0–10	6.7 ± 1.6	6.6 ± 1.9	6.2 ± 2.2	1.753	29.35	<0.001
Identity	0–10	3.7 ± 2.6	5.9 ± 2.3	5.3 ± 2.6	1.984	74.45	<0.001
Concern	0–10	5.9 ± 2.0	6.8 ± 1.9	5.1 ± 2.4	1.818	54.35	<0.000
Understanding	0–10	6.7 ± 1.4	7.9 ± 1.1	5.9 ± 2.4	1.564	125.27	<0.000
Emotional Response	0–10	3.4 ± 2.4	5.6 ± 2.2	5.8 ± 2.1	1.894	111.4	<0.000
Overall mean score		43.9 ± 9.9	53.8 ± 5.6	45.2 ± 11.5			<0.000

* *p*-value from repeated measures one-way ANOVA.

**Table 5 healthcare-11-00311-t005:** Association of patients’ clinical characteristics and illness perception at T2.

Clinical Characteristics	Illness Perception	*p*-Value	Illness Perception	*p*-Value
*Previous cardiac history*				
Angina				
Yes	1	0.54	1	0.131
No	1.211 (0.65, 2.23)		0.524 (0.23, 1.21)	
AMI				
Yes	1	0.001	1	0.002
No	2.372 (1.40, 4.00)		2.380 (1.46, 5.49)	
Angioplasty with stent				
Yes	1	0.61	1	0.004
No	1.177 (0.63, 2.20)		3.857 (1.55, 9.61)	
Open heart surgery				
Yes	1	0.002	1	0.002
No	2.119 (1.33, 3.39)		2.239 (1.27, 3.96)	
*Comorbidities*				
Diabetes				
Yes	1	0.013	1	0.914
No	0.564 (0.36, 0.87)		0.958 (0.45, 2.05)	
Peripheral vascular disease				
Yes	1	0.373	-	-
No	0.682 (0.23, 1.58)			
Stroke				
Yes	1	0.783	1	0.4
No	0.879 (0.35, 2.12)		3.342 (0.20, 55.4)	
*Cardiac Risk Factors*				
Current smoker				
Yes	1	0.092	1	0.275
No	1.560 (0.93, 2.62)		0.619 (0.62, 1.47)	
History of smoking				
Yes	1	0.052	1	0.52
No	1.612 (0.99, 2.61)		0.763 (0.34, 1.74)	
Hypercholesterolemia				
Yes	1	0.001	1	0.036
No	0.409 (0.25, 0.67)		0.417 (0.18, 0.94)	
Hypertension				
Yes	1	0.845	-	-
No	0.952 (0.58, 1.60)			
Sedentary Lifestyle				
Yes	1	0.496	-	-
No	0.847 (0.53, 1.40)			
Cardiologist care				
Yes	1	0.194	1	0.552
No	0.744 (0.47, 1.16)		1.259 (0.59, 2.68)	
Attend any CRP				
Yes	1	0.226	1	0.754
No	0.733 (0.44, 1.12)		0.894 (0.44, 1.80)	

## Data Availability

The data presented in this study are available on request from the corresponding author.

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
