# Peer review of "Changes in Perception of Illness during Cardiac Rehabilitation Programme among Patients with Acute Coronary Syndrome: A Longitudinal Study"

_healthcare, 2023, doi:10.3390/healthcare11030311_

Round 1

Reviewer 1 Report

Dear author

This article aims to determine changes in perception of illness with ACS patients during CRP to evaluate the association of the patient’s characteristics, and showed that the history of AMI, angioplasty intervention, and open-heart surgery were found to be associated with the perception of illness. That is very interesting, but I have several questions and there are comments that should be added as follows:

1)     Does ACS in this article include both of unstable angina (UAP); non-ST elevation AMI and ST elevation AMI?

2)     Does Angina mean stable angina? Is unstable Angina (UAP) included?

3)     In 2.3 participants section, the severity of AMI and cardiac function in participants should be shown, such as peak CK value, left ventricular ejection fraction (LVEF) of echocardiogram and rates of the presence of ST-elevation.

4)     Please added reasons why angioplasty intervention and open-heart surgery after AMI were not performed, such as high age, dementia and so on, to Material and Methods section.

5)     Could results of this study be changed If the severity or cardiac function after AMI were included in analysis?

Sincerely,

From reviewer.

Reviewer 2 Report

The article has merit and relevance, as it presents an instrument that is rarely used in our cardiac rehabilitation routine.

Suggestions:

- Add multicenter study in the title and methodology.

- In the limitations add that the instrument can be considered subjective in the patient's perception.

Round 2

Reviewer 1 Report

Dear author

Thank you very much for your comments and revisions. I think that the author can deal with all questions and comments, so this article deserves acceptable.

From reviewer.